# “OMICS” in Human Milk: Focus on Biological Effects on Bone Homeostasis

**DOI:** 10.3390/nu16223921

**Published:** 2024-11-17

**Authors:** Ilaria Farella, Gabriele D’Amato, Andrea Orellana-Manzano, Yaritza Segura, Rossella Vitale, Maria Lisa Clodoveo, Filomena Corbo, Maria Felicia Faienza

**Affiliations:** 1Department of Medicine and Surgery, LUM University, 70010 Casamassima, Italy; farella@lum.it; 2Neonatal Intensive Care Unit, Di Venere Hospital, 70012 Bari, Italy; gab59it@yahoo.it; 3Escuela Superior Politécnica del Litoral, ESPOL, Laboratorio Para Investigaciones Biomédicas, Facultad de Ciencias de la Vida (FCV), ESPOL Polytechnic University, Campus Gustavo Galindo Km 30.5 vía Perimetral, Guayaquil P.O. Box 09-01-5863, Ecuador; akorella@espol.edu.ec (A.O.-M.); ysegura@espol.edu.ec (Y.S.); 4Giovanni XXIII Pediatric Hospital, University of Bari “A. Moro”, 70124 Bari, Italy; r.vitale7@studenti.uniba.it; 5Interdisciplinary Department of Medicine, School of Medicine, University of Bari “A. Moro”, 70100 Bari, Italy; marialisa.clodoveo@uniba.it; 6Department of Pharmacy-Drug Sciences, University of Bari “A. Moro”, 70125 Bari, Italy; filomena.corbo@uniba.it; 7Pediatric Unit, Department of Precision and Regenerative Medicine and Ionian Area, University of Bari “A.Moro”, 70124 Bari, Italy

**Keywords:** human milk bioactive compounds, bone physiology, nutrigenomics

## Abstract

Human milk (HM) is a complex biofluid rich in nutrients and bioactive compounds essential for infant health. Recent advances in omics technologies—such as proteomics, metabolomics, and transcriptomics—have shed light on the influence of HM on bone development and health. This review discusses the impact of various HM components, including proteins, lipids, carbohydrates, and hormones, on bone metabolism and skeletal growth. Proteins like casein and whey promote calcium absorption and osteoblast differentiation, supporting bone mineralization. Long-chain polyunsaturated fatty acids like docosahexaenoic acid (DHA) contribute to bone health by modulating inflammatory pathways and regulating osteoclast activity. Additionally, human milk oligosaccharides (HMOs) act as prebiotics, improving gut health and calcium bioavailability while influencing bone mineralization. Hormones present in HM, such as insulin-like growth factor 1 (IGF-1), leptin, and adiponectin, have been linked to infant growth, body composition, and bone density. Research has shown that higher IGF-1 levels in breast milk are associated with increased weight gain, while leptin and adiponectin influence fat mass and bone metabolism. Emerging studies have also highlighted the role of microRNAs (miRNAs) in regulating key processes like adipogenesis and bone homeostasis. Furthermore, microbiome-focused techniques reveal HM’s role in establishing a balanced infant gut microbiota, indirectly influencing bone development by enhancing nutrient absorption. Although current findings are promising, comprehensive longitudinal studies integrating omics approaches are needed to fully understand the intricate relationships among maternal diet, HM composition, and infant bone health. Bridging these gaps could offer novel dietary strategies to optimize skeletal health during infancy, advancing early-life nutrition science.

## 1. Introduction

Human milk (HM) is widely acknowledged as the best nutritional source for neonates and infants in the first six months of life [1]. It is a complex and dynamic biological fluid that not only nourishes but also actively interacts with the infant’s immune system through a rich array of bioactive compounds. These include immunoglobulins, peptides, hormones, leukocytes, microorganisms, carbohydrates, milk fat globule membranes (MFGMs), and intracellular vesicles, which work both individually and in coordination to support and guide the infant’s growth and development. Moreover, HM functions as part of a broader ecological system encompassing the nursing mother, the infant, and their shared environment. Many of its components also likely play a role in providing personalized communication, helping the infant thrive within their unique cultural and environmental context [2]. As a dynamic biological system, HM is an ideal model for the application of “omics” sciences—genomics, proteomics, transcriptomics, and metabolomics—whose use in nutrition research, or foodomics, has rapidly advanced [3]. Omics sciences form the foundation of nutrigenetics and nutrigenomics, which offer insights into how genetic variations, such as single nucleotide polymorphisms, affect nutrient metabolism and how nutrients can, in turn, modulate gene expression through epigenetic or transcriptional changes [4]. These emerging fields aim to characterize diet–health interactions at a molecular level, identifying the specific nutrients responsible for observed health benefits and exploring how these components are metabolized within the body [5]. Leveraging advanced technologies like mass spectrometry, next-generation sequencing, and microarray analysis, researchers can analyze genome-wide cellular and molecular responses to HM, producing extensive datasets for bioinformatics analysis to derive functional insights [6]. Furthermore, omics technologies have greatly advanced our understanding of HM as a microbial ecosystem. Studies demonstrate that HM provides beneficial bacteria to the neonatal gut microbiota, supporting the maturation of both digestive and immune functions. By incorporating microbiome-focused techniques such as culturomics, this field allows for a comprehensive characterization of the HM microbiome’s composition and its interactions with other milk components [7]. Beyond the well-known nutritional benefits of HM, recent innovations in omics technologies have allowed for deeper insights into HM’s complex composition and its influence on infant bone homeostasis. These cutting-edge approaches reveal how HM components contribute to the regulation of bone mineralization and skeletal development. For instance, key nutrients like calcium, vitamin D, and proteins found in HM are essential for bone health, supporting osteoblast function and ensuring proper calcium metabolism [8]. Additionally, bioactive oligosaccharide compounds in HM (HMOs) indirectly promote bone physiology by maintaining a healthy gut microbiota, which, in turn, enhances the absorption of critical nutrients, including calcium and vitamin D [9]. Studies exploring the impact of HM on bone mineral content (BMC) and bone mineral density (BMD) have reported mixed findings. While some research points to a positive association between breastfeeding and increased bone mass during childhood and adolescence, other studies show no significant effect or even a potential negative impact [10]. Despite these conflicting results, the adaptive nature of HM remains clear, tailoring its composition to meet the evolving needs of the infant at each developmental stage.

This review seeks to provide a comprehensive overview of the current evidence on how omics technologies have enabled a better understanding of the role of HM in bone tissue homeostasis through the analysis of its biological components.

## 2. HM Composition: Role of Micro- and Macronutrients in Nourishing, Protecting, and Interconnecting Complex Information to the Infant

As a biological system, changes in the normal composition of HM can occur in response to the ecosystem (eco-homeorhesis), particularly across different geographic areas and cultures, and also in response to the lactation process (homeorhesis) [2]. Colostrum, the first breast fluid produced postpartum, is particularly enriched with immune factors such as secretory immunoglobulin A, lactoferrin, and leukocytes, offering immediate protection against pathogens and helping to establish immune defenses [11]. Growth factors in colostrum, like the epidermal growth factor, also play key roles in the development and maturation of the gastrointestinal system [12]. As lactation progresses, the composition of HM evolves to meet the infant’s biological needs. Mature HM continues to guide the growth and development of the immune system with a balanced mix of proteins, fats, and carbohydrates. All these vital compounds comprise the abundant presence of long-chain polyunsaturated fatty acids like DHA and are essential for the development of the central and peripheral nervous system and sense organs [13].

HM’s composition is influenced by various factors, including gestational age (in the case of neonatal preterm birth), maternal nutrition, genetic and physical characteristics, and stage of lactation [14]. In a full-term pregnancy, protein, fat, and carbohydrate levels fluctuate during lactation, with fat increasing and protein decreasing over time. The relationship between gestational age and macronutrient levels in HM is not fully understood. Some research indicates no significant differences in macronutrient levels between very preterm and preterm milk during the first week of lactation [14,15]. However, other studies report that protein levels are higher in preterm and very preterm milk compared to term milk, particularly during the early days and throughout the first two months of lactation [16]. Over time, protein concentrations tend to decrease in both preterm and term milk, with fat content initially higher in preterm milk but lower after the first few weeks. Carbohydrate levels have shown mixed results, with some studies reporting higher levels in term milk, while others find no significant differences [17,18,19].

### 2.1. Micronutrients

HM contains all the essential micronutrients (vitamins and minerals) that infants require. Concentrations of some of these are influenced by maternal status and diet or supplementation (e.g., thiamine, riboflavin, niacin, biotin, pantothenic acid, pyridoxine, cyanocobalamin, vitamin C, vitamin A, vitamin D, vitamin E, vitamin K, choline, iodine, and selenium); others are independent of maternal intake (e.g., folate, calcium, iron, copper, zinc, sodium, chloride, potassium, phosphorus, magnesium, manganese, fluoride, chromium, and molybdenum) [20,21]. There is a circadian variation in milk micronutrient concentrations, which is related to food consumption throughout the day and also to milk lipid content [22]. Specifically, the calcium content in HM is typically around 261 mg/L, with a slight decrease as lactation progresses. This concentration remains relatively stable across different populations, though maternal health conditions can influence levels. Phosphorus content in HM is generally consistent and plays a crucial role in supporting the infant’s bone mineralization and overall growth. The average phosphorus concentration in breast milk is approximately 140 mg/L. This level remains relatively stable throughout lactation, similar to calcium. Phosphorus, in combination with calcium, is vital for the formation of strong bones and teeth in infants. Vitamin D content in breast milk, on the other hand, varies more significantly, with an average concentration of 58 IU/L. Factors such as maternal vitamin D supplementation, geographic location, and exposure to sunlight can greatly affect these levels. Vitamin D3 (25OHD3) is the most prevalent form found in breast milk. While calcium and phosphorus levels generally meet the nutritional needs of infants, vitamin D levels are often insufficient, and supplementation may be necessary to ensure optimal infant health [23]. Beyond calcium, phosphorus, and vitamin D, HM includes a range of other vitamins and minerals that support various developmental processes in infants. Vitamin A supports immune function and vision, vitamin C is essential for collagen synthesis and cellular protection, and vitamin E acts as a key antioxidant, protecting cells from oxidative damage. These vitamins, together with trace minerals like zinc and copper, contribute to diverse physiological processes, including immune system support, antioxidant defenses, and enzymatic reactions crucial for development. Each component of HM, influenced by maternal factors or maintained independently, reflects an evolved composition that adapts to support the growth and health of infants [22].

### 2.2. Macronutrients

HM contains an array of macronutrients that nourish, protect, and have a role in communication between the mother and infant [24].

Proteins. Recent advancements in “omics” approaches have led to a much wider understanding of the numerous proteins in HM. Indeed, proteomics analyses have revealed that HM contains hundreds of unique proteins with an array of functional characteristics, including enhancing nutrient absorption (e.g., bile salt–stimulated lipase), defending against pathogens (e.g., lactoferrin, lysozyme, immunoglobulins), shaping the infant’s immune system (e.g., cytokines), and guiding the development of the gastrointestinal tract (e.g., epidermal growth factor and insulin-like growth factor 1) [18,19]. They can be divided into three groups: caseins, whey proteins, and mucin proteins. Caseins, a family of proteins mainly composed of α-caseins (αs1 and αs2 caseins), β-caseins, and κ-caseins, are organized into micelles and have multiple biological functions in newborns, especially in transporting calcium phosphate from the mother to the infants. HM has a lower casein content compared to other species, reflecting the slow growth rate of human infants [25]. Whey proteins, which include α-lactalbumin, lactoferrin, immunoglobulins, serum albumin, and lysozyme, are dissolved in the milk solution and are involved in immune regulation and antimicrobial protection. Mucins are located in the milk fat globule membrane and contribute to the structural and protective qualities of the milk [18].

In addition to proteins, non-protein nitrogen makes up about 25% of the nitrogen in HM [25]. This includes compounds like urea, creatinine, nucleotides, free amino acids, and peptides, many of which have crucial roles in metabolic regulation, gastrointestinal development, and immune function [26].

The full range of known bioactive peptides in HM (as well as the milk of other species) is cataloged in the milk bioactive peptide database [27], although the extent to which human milk proteins and peptides survive within the infant’s gastrointestinal tract and have the potential to exert their bioactivities at sites of gastrointestinal action remains largely unknown.

Lipids. Lipids in HM, comprising about 50–60% of the total energy, are crucial for infant growth, brain development, and nerve function [28]. The fat content includes triacylglycerols, phospholipids, and cholesterol, all of which are essential for cell membrane structure and function. HM fat globules play a key role in facilitating digestion. Studies suggest that smaller milk fat globule sizes in HM enhance fat absorption compared to bovine milk [29]. Breast milk contains more than 200 fatty acids, with oleic acid being the most abundant, contributing around 30–40 g per 100 g of fat [28]. Approximately 17% of the fatty acids in breast milk are synthesized within the mammary gland (de novo synthesis) [28]. Long-chain polyunsaturated fatty acids, such as those with more than 20 carbon atoms, make up around 2% of the total fatty acids [19]. The positioning of fatty acids on the glycerol backbone affects their absorption, with palmitic acid at the sn-2 position being absorbed more efficiently [30]. This specific arrangement is not replicated in many infant formulas and is known to impact on the infant’s lipid profile, including cholesterol levels. In addition to their role in energy provision, short-chain fatty acids in breast milk are crucial for gastrointestinal development, and sphingomyelins, present in the fat globule membrane, are essential for the myelination of the central nervous system, particularly in low-birth-weight infants [31]. Moreover, breast milk lipids have demonstrated antibacterial properties, capable of inactivating pathogens like Group B streptococcus, which offers an additional layer of protection against infections at mucosal surfaces [32,33].

Carbohydrates. Lactose is the primary carbohydrate in HM, remaining constant during lactation. It aids in energy supply and mineral absorption, particularly calcium. This occurs as gut bacteria convert lactose into lactic acid, lowering pH and enhancing calcium solubility. Lactose is not broken down significantly in the upper gastrointestinal tract but is instead processed in the lower portions of the intestine [34]. HM also contains a large quantity of HMOs, complex sugars made up of five monosaccharides. These oligosaccharides vary in structure, with over 200 types identified [35]. HMOs are produced based on specific genes, such as the Secretor and Lewis blood group genes. They play a key role as prebiotics, supporting beneficial gut bacteria and modulating the infant’s immune system. Additionally, HMOs act as receptor decoys, preventing pathogens from binding to intestinal cells and thereby reducing infections [36]. The concentration of HMOs is highest in colostrum and decreases as milk matures. Despite advances, the chemical synthesis of HMOs remains costly and challenging, though some, like 2′-fucosyllactose (2′-FL) and lacto-N-neotetraose (LNnT), have been added to infant formula as novel ingredients. However, clinical trials have yet to provide definitive results regarding the benefits of HMO-fortified formulas.

Immunoglobulins. Immunoglobulins, especially secretory IgA, are abundant in breast milk, particularly in the early stages of lactation, and provide essential immune protection to the infant as their own immune system develops [37]. SIgA is the most prevalent form, followed by IgG, and these antibodies help protect the infant from pathogens. As the infant’s immune system becomes more independent, the concentration of antibodies in breast milk decreases. This decline also corresponds with the reduced permeability of the infant’s gut to macromolecules, limiting the absorption of larger proteins over time. In early life, breast milk plays a critical role in protecting mucosal surfaces because newborns have only trace amounts of SIgA and SIgM in their secretions. Studies show that breastfed infants have detectable IgA in their feces within two days of birth, while formula-fed infants take up to a month to develop similar levels [38]. These antibodies are produced as a response to maternal exposure to antigens, particularly through mucosa-associated lymphoid tissue (MALT) and the bronchomammary pathway [39]. This means that maternal immunizations or infections during the perinatal period shape the specific antibodies present in breast milk, such as those protecting against *Neisseria meningitidis* following vaccination [40]. SIgA is particularly resistant to digestion and is thought to be the primary defense against mucosal pathogens. It works by neutralizing pathogens, preventing them from adhering to epithelial cells, and neutralizing toxins. SIgA in breast milk has been shown to protect against a wide range of pathogens, including *Vibrio cholerae*, *Campylobacter*, *Shigella*, *Giardia lamblia*, and respiratory infections [41,42]. The presence of these antibodies in breast milk provides crucial protection to infants against both enteric and respiratory infections. Breast milk also contains antibodies against Group B Streptococcus, providing an additional layer of protection. SIgA antibodies may prevent GBS from adhering to epithelial cells, reducing the risk of infection, especially in preterm infants [43].

Bioactive compounds. Breast milk contains numerous compounds that manifest biological properties, such as antimicrobial activity, modulation of the immune system, antioxidant function, and enzymatic regulation. These bioactive substances include hormones, cytokines, nucleotides, immunoglobulins, lactoferrin, lysozyme, and cells and bacteria. Many peptides derived from caseins and whey proteins are liberated through biological processes, such as digestion or fermentation, through gut microbiota. These compounds act both locally in the gastrointestinal tract and systemically once absorbed into the bloodstream [27]. Their function is to respond to the infant’s needs, offering defenses against pathogens and supporting gastrointestinal maturation, brain development, and bone physiology.

## 3. Factors Influencing the Composition of HM

Multiple maternal factors have been associated with variations in HM composition, such as diet, maternal age, ethnicity, weight gain during pregnancy, child’s birth weight, and smoking [44,45,46,47]. The relationship between maternal diet and breast milk composition is intricate. A systematic review by Petersohn et al. found that the maternal consumption of fat, monounsaturated fatty acids, and cholesterol was positively associated with the energy and fat content in breast milk, while docosahexaenoic acid intake correlated with higher docosahexaenoic acid levels [46]. However, no strong associations were found for other nutrients, such as proteins, carbohydrates, vitamins, and minerals. The study highlighted the need for more standardized research due to inconsistencies in design and methodology. It also underlined the importance of fish and oil intake for improving docosahexaenoic acid and polyunsaturated fatty acids levels in milk but called for further exploration into other nutrients’ effects. An analysis of breast milk composition across seven countries reveals that while it remains largely consistent across ethnicities, fat content displays the most variation. Interestingly, the degree of variation within mothers of the same ethnicity is as significant as between different ethnic groups [48]. Maternal mineral supplementation has shown clear effects, such as iodine increasing HM iodine concentrations [49,50]. Similarly, selenium supplementation improves HM selenium levels [51,52], though zinc and iron content in HM are not largely affected by dietary intake due to tightly regulated homeostasis [53,54]. Vitamin supplementation is particularly effective in elevating certain micronutrients in HM. For example, B-vitamin supplementation rapidly increases its concentration in HM [55], while higher doses of vitamin B12 [56,57] and choline supplementation result in elevated HM levels [58,59]. Lutein [60,61] and vitamin A concentrations in HM also benefit from higher maternal intake [62]. Vitamin D supplementation in doses higher than the current recommendations may be required to adequately transfer vitamin D to the breastfed infant [63,64]. On the other hand, galactagogues, substances believed to increase milk production, have garnered interest, but the evidence remains anecdotal, and the quality of research is low. The herbal galactagogue fenugreek has shown efficacy against placebo in clinical trials [65], but more robust studies are needed. Maternal body composition, such as body mass index (BMI), can influence the lipid profile of HM. Overweight women tend to have higher saturated fat levels in their HM [66], while leaner women show higher polyunsaturated fatty acid levels [67].

Additionally, maternal body fat has been positively associated with HM leptin levels, which, in turn, correlates with infant weight gain [68,69]. Maternal obstetric history, including parity and mode of delivery, can also impact HM composition. Multiparous women tend to have higher lipid content in their milk [70], while cesarean delivery has been associated with increased choline [71] and iodine levels in HM [72]. Additionally, the concentration of fat in HM increases in mothers who give birth to infants with either low or high birth weights, while protein and carbohydrate levels remain largely unaffected [73]. Smoking negatively impacts breast milk by reducing its protective properties and altering its composition, potentially harming infant health. Nicotine concentrations in the breast milk of smokers are found to be three times higher than in their plasma. Additionally, smoking decreases milk volume and shortens the breastfeeding period [74]. In addition to environmental and lifestyle factors, genetic variations significantly influence the nutritional content of HM. These variations affect the secretion of important nutrients such as zinc, iodine, and fatty acids, which are essential for infant growth and development. For example, mutations in the SLC30A2/ZnT2 gene can result in low levels of zinc in breast milk, impacting the infant’s zinc intake and requiring supplementation. Similarly, variations in the ABCG2 gene affect the transport of riboflavin and other nutrients, influencing their concentration in breast milk. Variations in the FADS1 and FADS2 genes influence the levels of polyunsaturated fatty acids like DHA and arachidonic acid, which are critical for infant brain development [75]. Furthermore, as previously discussed, HM composition is closely tied to the lactational stage, from colostrum to mature milk (from 16 days onward). These stages correlate with mammary gland maturity, with early colostrum showing the highest protein concentration, particularly immune-modulating proteins, as the mammary gland continues to develop. During this phase, tight junction closures in the mammary epithelium help regulate Na+ and K+ ions, essential for milk’s protective functions. Protein synthesis increases during the transitional stage, eventually giving way to fat synthesis in mature milk.

HM reaches relative compositional stability between 2 and 12 weeks postpartum [76]. Protein concentration generally starts high in colostrum, then declines to stabilize at about 10–20 g/L in mature milk. Research suggests that this adaptive shift from protein dominance to fat synthesis aligns with the infant’s need for more energy-dense nutrition as the gut microbiome diversifies. The evolving protein profile in human milk supports immune development, shifting from initial pathogen defense (via IgA and IgM) to immune system support as the milk increases in IgG content [77]. Additional factors like diurnal fluctuations and the influence of mammary gland mediators also affect milk’s proteome, though their impact on protein and post-translational modifications is less explored [78] (Figure 1).

Several maternal factors influence the composition of HM, including maternal diet, vitamin and mineral supplementation, BMI, body composition, obstetric history (parity and mode of delivery), smoking, and genetic variations. Maternal diet primarily affects levels of fats, DHA, cholesterol, and leptin and is also influenced by obstetric history. Vitamin (B12, D, A, lutein) and mineral (iodine, selenium) supplementation increase the respective concentrations in HM. Genetic variations in the SLC30A2/ZnT2, ABCG2, FADS1, and FADS2 genes affect the secretion and levels of zinc, riboflavin, and polyunsaturated fatty acids (DHA and arachidonic acid). Additionally, smoking decreases milk volume and reduces its protective properties, while maternal body composition and obstetric history further modulate the lipid and protein content in milk. The lactational stage, from colostrum to mature milk (from day 16 onward), significantly impacts HM composition. Early-stage colostrum is protein-rich, especially immune-modulating proteins, while fat synthesis becomes prominent as mature milk forms. Diurnal fluctuations and lactation stages also influence the HM proteome, as protein concentration declines and stabilizes in mature milk, aligning with the infant’s energy needs and gut microbiome diversification. 

## 4. The Role of Maternal and Early-Life Nutrition in Bone Development

The early years of life are critical for skeletal growth and development, with foundational theories such as Barker’s hypothesis supporting the concept that early-life factors can program long-term health outcomes, including bone health [79,80,81,82]. Bone formation begins during the prenatal phase, where cells differentiate into chondrocytes and osteoblasts, with the majority of bone accretion occurring in the third trimester [83]. This period of rapid growth extends into infancy, with continued development up to peak bone mass around age 20, influenced by skeletal site, sex, and other factors [83,84]. Genetic and epigenetic factors, along with nutrient availability, play a substantial role in shaping peak bone mass, which is an essential determinant in reducing osteoporosis risk later in life [85,86]. Vitamin D and calcium are often highlighted as crucial dietary components for bone health. Studies indicate that maternal or direct vitamin D supplementation is effective in improving serum 25OHD levels in both mothers and infants, with high doses significantly reducing vitamin D deficiency in both populations [87,88,89]. After birth, childhood factors, like a balanced diet rich in calcium and vitamin D, along with regular physical activity, significantly contribute to healthy bone development and serve as a foundation for osteoporosis prevention later in life [90]. However, in the first six months post-term, preterm infants small for gestational age, despite receiving appropriate nutritional intervention through breastfeeding, show lower bone accretion compared to those appropriate for gestational age, suggesting that prenatal conditions may remain compromised [91].

Research suggests several mechanisms by which maternal intake of specific nutrients may influence offspring bone health. Higher maternal protein intake is linked to greater bone mineral accrual and increased levels of IGF-1, which is associated with higher BMD [88,92]. Conversely, saturated fats have shown an inverse relationship with bone density in adults [93], and animal studies suggest that high fat intake may decrease calcium absorption, potentially adversely affecting offspring bone development [94]. Magnesium deficiency could impact bone health by altering calcium homeostasis and modulating bone cell activity, diminishing osteoblast function, and enhancing osteoclast activity, as shown in both in vitro and in vivo studies [95]. Vitamin D also plays a critical role; in fact, a low maternal vitamin D may impair fetal skeletal growth and mineralization due to inadequate placental calcium transport [96,97,98]. However, some studies suggest compensatory mechanisms in infants born to mothers with low vitamin D that help maintain skeletal growth [99,100].

Long-chain polyunsaturated fatty acids in maternal diets are also linked to positive bone outcomes, possibly due to the osteoblast-promoting and osteoclast-inhibiting effects of eicosanoids derived from essential fatty acids [101]. Vitamin A is crucial in embryonic skeletal formation and modulates epigenetic processes, yet high intake from supplements may be associated with bone loss [102]. Additionally, folate and vitamin B12 may influence bone health through epigenetic mechanisms as DNA methylation contributors, with potential direct effects on osteoblast activity and homocysteine metabolism [103].

Numerous studies have compared BMD and BMC in both term and preterm infants under different feeding regimens [104,105,106,107]. HM is associated with lower total body BMC compared to formula, possibly due to its lower vitamin D and phosphorus levels. Although vitamin D in human milk is low and phosphorus levels decrease with prolonged lactation [108,109,110,111,112], the effect of these nutrients on BMD is not consistent across studies. For instance, it has been demonstrated that vitamin D supplementation affects serum 25-hydroxyvitamin D levels and may prevent a decline in BMC initially, but these effects diminish over time [86,89].

## 5. Omics in Colostrum and Mature Milk

Colostrum and mature milk provide essential nutrients and bioactive compounds that adapt throughout lactation to meet the infant’s needs. Advances in omics technologies have revealed key changes in milk composition, especially in minerals and proteins, which support infant growth, immune response, and bone health (Figure 2).

### 5.1. Minerals

Minerals are indispensable nutrients that play essential roles in promoting and sustaining bone growth. A recent study analyzed a total of 200 breast milk samples from seven cities in China to detect mineral and trace elements using inductively coupled plasma mass spectrometry [113]. Three distinct mineral patterns in human milk were identified by using inductively coupled plasma mass spectrometry (ICP-MS): Cluster I, characterized by the highest levels of potassium, magnesium, and calcium and the lowest levels of copper, zinc, manganese, and selenium; Cluster II, with the most abundant levels of sodium, iron, zinc, manganese, and selenium; and Cluster III, marked by the lowest levels of sodium, potassium, magnesium, iron, and calcium. The authors found that a specific Cluster I mineral pattern is associated with greater infant growth. Compared with other clusters, HM samples of Cluster I showed the most evident variation of metabolites of arachidonic acid (ARA) and the nicotinate and nicotinamide metabolism pathway.

Studies have demonstrated that ARA plays an important role in regulating bone formation and mineral metabolism during the growth and development of infants. Firstly, ARA is the precursor for prostaglandin E2 (PGE2), a potent bone-resorbing agent that stimulates the synthesis of insulin-like growth factor 1 (IGF-1) and insulin-like growth factor binding protein 5 (IGFBP-5), which promote osteoblast differentiation [114,115]. In addition, ARA and PGE2 interact with the 1,25 (OH)2D3 pathway, which regulates calcium absorption, and thus, in Cluster I, free ARA appears to be metabolized to a greater extent, thus undergoing conversion to downstream products and resulting in lower residual ARA content. These differences observed in Cluster I could explain the better impact on osteoblasts and chondrocytes and, consequently, on growth.

### 5.2. Proteins

Proteins in HM play a central role in supporting infant growth, supplying essential amino acids that are readily accessible after digestion. These proteins, along with the peptides they produce, also aid in growth by enhancing both nutrient absorption and digestion efficiency. β-casein generates casein phosphopeptides in the mammary gland and during infant digestion, which act as chelating agents that bind minerals like calcium, zinc, and iron to facilitate their absorption [116,117]. β-lactalbumin, the most abundant whey protein, includes binding sites for calcium and zinc, further aiding in mineral uptake. Recent studies have shown that casein-derived bioactive peptides promote osteoblast proliferation and differentiation, underscoring their importance in bone formation and development [118]. The study by Wang et al. used data-independent acquisition proteomics to explore casein proteins in human milk from Korean and Han Chinese mothers, focusing on protein composition differences between these groups. Researchers identified 535 proteins in the casein fraction, with 39 showing differential expression—10 upregulated and 29 downregulated. The functional analysis highlighted pathways associated with carbohydrate metabolism and immune response, with a specific enrichment in *Staphylococcus aureus* infection pathways and blood microparticle functions. The *Staphylococcus aureus* pathway reflects that certain proteins in casein are involved in immune pathways typically activated in response to such bacteria. This suggests that casein proteins in human milk may play a key role in immune defense, possibly reflecting evolutionary adaptations for infant immunity in different populations. Additionally, findings align with earlier research indicating that bioactive peptides derived from casein support immune functions and promote bone health by encouraging osteoblast proliferation, differentiation, and mineralization [119].

### 5.3. Carbohydrates

Carbohydrates are another critical component of HM, with HMOs being particularly noteworthy due to their multifaceted role in regulating infant gut microbiota. They are synthesized by glycosyltransferases, enzymes that combine five core monosaccharides into more than 150 structurally diverse oligosaccharides [120]. HMOs are grouped into three primary classes: neutral non-fucosylated, neutral fucosylated, and sialylated, with notable compounds like 2-fucosyllactose (2-FL) and lacto-N-tetraose (LNnT) contributing to their functional diversity [121,122].

HMOs that are not digested by infants support the growth of beneficial bacteria such as *Bifidobacterium*, a key genus in the infant microbiome that can utilize HMOs for energy through specific glycoside hydrolases, producing short-chain fatty acids that improve gut health by promoting mucin secretion and regulating immune responses [123,124]. Furthermore, these bacteria that contribute to a healthier gut environment can, in turn, facilitate the absorption of key nutrients like calcium, enhancing its bioavailability and supporting bone mineralization [125,126]. Additionally, by influencing the immune system, HMOs may help to regulate systemic inflammation, a known factor that can negatively affect bone health through increased bone resorption [127].

Over the past two decades, metabolomics has emerged as a powerful tool for studying nutrients like HMOs. In a pioneering study, HMO profiles were associated with different Lewis phenotypes, and it was found that certain oligosaccharides, like 2′-FL, had protective effects against *Escherichia coli*-induced diarrhea [113]. Thurl et al. further confirmed that HMO expression correlates with maternal phenotype, noting that 2′-FL concentrations peak shortly postpartum before decreasing by 50% by three months [128].

Other studies revealed similar trends, such as Gabrielli et al., who found that total HMO levels decline within the first month but vary by lactation stage [129]. The impact of maternal factors, like weight, on HMO composition has also been explored. Saben et al. found elevated concentrations of lacto-N-tetrose (LNT) and LNnT in overweight mothers, suggesting potential links to maternal BMI [130]. Sundelkilde et al. noted that HMO profiles shift significantly between preterm and term births, with compounds like 3-FL, lactose, and glucose increasing with milk maturation [131].

HMO composition also appears to vary with both geographical factors and maternal phenotype, with studies showing distinct HMO patterns by location and secretor status [132,133].

### 5.4. Growth Factors and Hormones

In vivo studies suggest that breast milk positively influences BMD and BMC during the lactation period and beyond, potentially due to the presence of bioactive components such as growth factors, like IGF-1 and TGF-β, as revealed by proteomic and metabolomic analyses [134]. IGF-1 and other hormones, such as leptin, ghrelin, adiponectin, and insulin, appear to play crucial roles in infant growth and body composition [135].

For instance, higher levels of IGF-1 in breast milk have been associated with increased weight gain in infants, as demonstrated by studies integrating multi-omics approaches. Additionally, breast milk leptin has been inversely correlated with infant adiposity and trunk fat at six months of age [136,137,138]. Ghrelin, which has demonstrated a positive relationship with weight gain during the first two months of lactation [139], and adiponectin, which has been linked to both lower fat-free mass and increased fat mass during the first year of life [140], are other hormones whose molecular mechanisms and interactions with bone metabolism could be more comprehensively understood using omics-based studies. Although the specific impact of these hormones on bone metabolism has not been directly demonstrated through omics profiling, their established roles in regulating bone health suggest that they may exert indirect effects when present in breast milk [141].

For example, leptin can influence bone formation by modulating osteoblast and osteoclast activity through its central action on the hypothalamus [142], as revealed by proteomic data. Similarly, ghrelin has been reported to promote bone growth by directly stimulating osteoblast differentiation and increasing IGF-1 secretion [143], highlighting the potential role of these molecules in skeletal development. Furthermore, adiponectin, known for its anti-inflammatory properties, enhances insulin sensitivity by promoting glucose uptake in muscle and increasing fatty acid oxidation through AMP-activated protein kinase pathways. This dual role of adiponectin in supporting insulin response and directly enhancing osteoblast proliferation and mineralization contributes to improved bone mineral density (BMD) [88]. Omics-based research, particularly using proteomics and transcriptomics, could provide deeper insights into how these hormones regulate bone homeostasis at the molecular level during this critical stage of skeletal development, which often coincides with exclusive breastfeeding. Despite the promising evidence, the precise molecular mechanisms and causal factors underlying this influence remain elusive, representing a fertile area for future research leveraging integrative omics approaches. Although there are currently no clinical trials in infants that directly demonstrate the impact of these hormones on bone health using omics methodologies, emerging evidence suggests that they may have an indirect role through several mechanisms that could be explored through multi-omics studies in the future.

### 5.5. MicroRNAs

MicroRNAs (miRNAs), small RNA molecules that regulate gene expression in cells, are demonstrated in colostrum, pre-colostrum, and transitional milk and have been shown to play essential roles in bone homeostasis and infant growth. Omics technologies have analyzed how miRNAs respond to maternal diet during gestation and lactation, which modifies both the composition of colostrum and its microbiota, influencing not only the infant’s gut microbiota but also playing a crucial role in regulating key cellular processes, such as cell proliferation, differentiation, and tissue repair [144]. For instance, a study of 60 Spanish mothers showed that miRNA levels in milk vary according to maternal diet and protein intake, with higher amounts of miRNAs related to cell growth and proliferation in mothers consuming predominantly vegetable proteins [145].

Integrative multi-omics data were used to analyze the association between miRNAs, maternal dietary nutrients, and the 16S rRNA gene of the infant microbiota and breast milk. Specifically, 10 miRNAs—such as those from miR-378 and the miR-320 family—were positively associated with the infant BMI Z-score at 6 months of age in infants who received early breastfeeding [146]. These miRNAs are also involved in adipogenesis, promoting adipocyte differentiation of skeletal and bone marrow mesenchymal stem cells by targeting the Runt-related transcription factor 2 gene [146].

Omics technologies have demonstrated that the composition of HM plays a crucial role in influencing bone development and homeostasis in infants. In metabolomics and epigenomics, maternal micronutrients, such as vitamin D and mineral concentrations, are associated with bone development and bone mass parameters in infants, while hormones and metabolic markers like glucose, TNF-α, and leptin correlate with growth during the first six months of life. Transcriptomics has identified maternal miRNAs that are linked to infants’ BMIs, suggesting their role in early growth regulation. Proteomics has shown that proteins such as immunoglobulins and casein in HM are crucial for osteoblast function, promoting bone growth in infants. Microbiomics has revealed how growth factors and miRNAs are involved in the regulation of the microbiome in both mothers and infants, further contributing to bone homeostasis. Together, these findings highlight the complex interplay between HM composition and bone health during infancy.

## 6. Conclusions

Early-life nutrition, particularly through HM, plays a foundational role in infant skeletal growth and bone health. Advances in omics technologies have clarified the essential roles of HM components—such as proteins, hormones, lipids, micronutrients, and microRNAs—in bone development and metabolism. Proteins like casein and bioactive peptides aid in calcium transport and osteoblast function, while growth factors like IGF-1 and hormones like leptin influence bone density and mineralization by modulating osteoblast and osteoclast activity. HMOs support bone health indirectly by fostering a beneficial gut microbiota that enhances nutrient bioavailability and may reduce inflammation, which could otherwise negatively affect bone remodeling.

HM micronutrients, including vitamin D and calcium, are vital to bone mineralization processes, while variations in these nutrients—due to maternal diet or lactation stage—can significantly impact bone accretion. MicroRNAs contribute to the regulation of adipogenesis and bone homeostasis by influencing early adipocyte and osteoblast differentiation, with certain microRNAs correlating with infant BMI and growth markers.

Despite these advancements, current research on HM’s effects on bone health has limitations. Many studies lack a comprehensive range of markers to reflect all aspects of osteogenic differentiation or fail to account for maternal serum levels of these nutrients prior to intervention. Moreover, the complexity of interactions between these bioactive compounds and infant bone metabolism suggests the need for extensive longitudinal studies spanning preconception and gestational periods to fully elucidate these mechanisms. Additionally, limitations involving sample selection and inclusion criteria may not accurately reflect the impact of exclusive breastfeeding on bone health.

Future research integrating omics methodologies in longitudinal studies could provide a clearer picture of how maternal nutrition and HM composition influence infant bone health. Such research could inform nutritional guidelines for nursing mothers, optimize early-life nutrition strategies, and promote bone health from infancy through to later life, ultimately reducing the risk of bone-related disorders.

## Figures and Tables

**Figure 1 nutrients-16-03921-f001:**
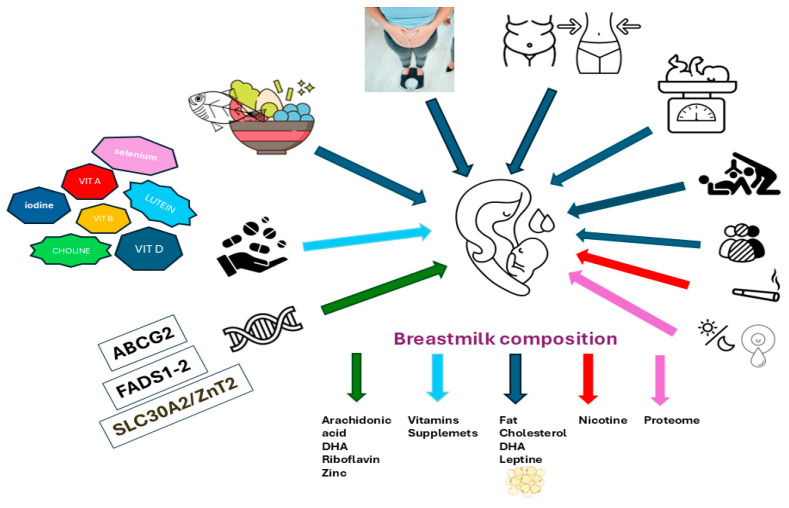
Factors influencing the composition of human milk.

**Figure 2 nutrients-16-03921-f002:**
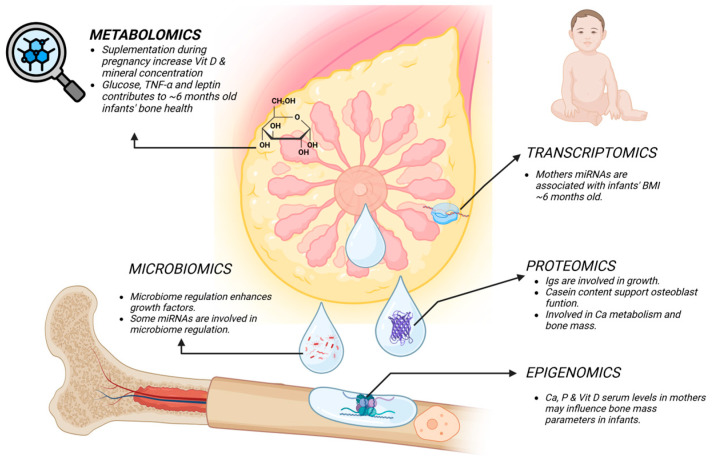
“Omics” in HM: influence on bone homeostasis. Legend: BMI, body mass index; Ca, calcium; Igs, immunoglobulins; miRNAs, microRNAs; P, phosphorus; TNF-α, tumor necrosis factor-alpha; Vit D, vitamin D; HM, human milk.

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
