# Peer review of "“OMICS” in Human Milk: Focus on Biological Effects on Bone Homeostasis"

_nutrients, 2024, doi:10.3390/nu16223921_

Round 1

Reviewer 1 Report

Comments and Suggestions for Authors

The authors of the manuscript described the importance of human milk in maintaining homoeostasis of the skeletal system in infancy. The manuscript is well-edited, based on current literature. The content of the manuscript corresponds to the title presented. The only reservations are two errors in the text. The first one concerns the citation of position 113, which appears in lines 364 and 373, in my opinion it should refer to the citation from line 373. The second error concerns chapter 5.1, where the authors describe the content of minerals in breast milk and the entry made may mislead the reader, because it is written, I quote, "A recent study analyzed a total of 200 breast milk samples from seven cities in China to detect mineral and trace elements [113]. Untargeted HM metabolomic profiles were determined using high performance liquid chromatography–tandem mass spectrometry (HPLC–MS/MS). Three HM mineral patterns were identified: cluster I characterized as the highest levels of potassium, magnesium and calcium, and the lowest levels of copper, zinc, manganese and selenium; cluster II characterized as the most abundant sodium, iron, zinc, manganese and selenium; cluster III characterized as the lowestlevels of sodium, potassium, magnesium, iron and calcium." In this entry Untargeted HM metabolomic profiles was determined using high performance liquid chromatography–tandem mass spectrometry (HPLC–MS/MS) is unnecessary and does not refer to minerals whose determination was made on ICP-MS. And the metabolic profile was indeed analyzed on UHPLC - Ms/Ms but it concerned other biological substances not related to this separation.

Author Response

The authors of the manuscript described the importance of human milk in maintaining homoeostasis of the skeletal system in infancy. The manuscript is well-edited, based on current literature. The content of the manuscript corresponds to the title presented. The only reservations are two errors in the text. The first one concerns the citation of position 113, which appears in lines 364 and 373, in my opinion it should refer to the citation from line 373. The second error concerns chapter 5.1, where the authors describe the content of minerals in breast milk and the entry made may mislead the reader, because it is written, I quote, "A recent study analyzed a total of 200 breast milk samples from seven cities in China to detect mineral and trace elements [113]. Untargeted HM metabolomic profiles were determined using high performance liquid chromatography–tandem mass spectrometry (HPLC–MS/MS). Three HM mineral patterns were identified: cluster I characterized as the highest levels of potassium, magnesium and calcium, and the lowest levels of copper, zinc, manganese and selenium; cluster II characterized as the most abundant sodium, iron, zinc, manganese and selenium; cluster III characterized as the lowest levels of sodium, potassium, magnesium, iron and calcium." In this entry Untargeted HM metabolomic profiles was determined using high performance liquid chromatography–tandem mass spectrometry (HPLC–MS/MS) is unnecessary and does not refer to minerals whose determination was made on ICP-MS. And the metabolic profile was indeed analyzed on UHPLC - Ms/Ms but it concerned other biological substances not related to this separation.

Dear Reviewer,

We would like to express our sincere gratitude for your comments and constructive suggestions. We have carefully reviewed each point raised and have made the requested revisions. Below, we address each of your observations.

  1. Correction of Bibliographic Reference

We apologize for the mistake. We have updated the manuscript with more appropriate bibliographic references, as per your suggestion. These citations provide more specific and accurate support for the content discussed.

  1. Rephrasing of the Section on Breast Milk Minerals and Use of ICP-MS Method

We have clarified the relevant paragraph to avoid any potential confusion regarding the analysis method used. As suggested, we removed the unnecessary reference to "untargeted HM metabolomic profiles" and specified that the mineral analysis was conducted using inductively coupled plasma mass spectrometry (ICP-MS).

Reviewer 2 Report

Comments and Suggestions for Authors

The article titled "OMICS in Human Milk: Focus on Biological Effects on Bone Homeostasis" explores the role of human milk (HM) in infant bone health through the lens of omics technologies, including proteomics, metabolomics, and transcriptomics. It discusses how specific HM components like proteins, lipids, hormones, and oligosaccharides contribute to bone metabolism, calcium absorption, and osteoblast differentiation, supporting infant skeletal growth. The article also highlights the role of maternal diet, genetic factors, and the infant's microbiome in shaping HM's composition and, consequently, its impact on bone health. Despite significant findings, the authors emphasize the need for longitudinal studies to fully understand the complex interactions between HM components and bone development, aiming to optimize early-life nutrition for better skeletal outcomes.

Below are some necessary changes.

Apply italics to all scientific names throughout the text. 

Line 40 – The keywords should be different from those used in the title, and it is recommended that they are not present in the abstract. 

Line 47 – "Microbes" or "microorganisms"? 

Lines 104-106 – Does the age of the mother influence the nutrient quality of HM? 

Lines 120-121 – Change vitamin B6 and vitamin B12 to pyridoxine and cyanocobalamin, respectively, to be consistent with the names used for other B-complex vitamins. 

Section 2.1 – The authors addressed the importance of calcium, potassium, and vitamin D. What about the importance of other vitamins and minerals? Or are they not significant? 

Lines 315-316 – Does the legend refer to Figure 1? 

Line 366 – "...essential nutrients and bioactive compounds..." Up to this point, the authors have presented macro- and micronutrients but have not introduced bioactive compounds. It would be appropriate to include a section on this class of components. 

Line 473 – Is there any relationship between adiponectin and insulin response? 

Figure 2 should be inserted right after it is mentioned in the text. 

It is necessary to include: Supplementary Materials, Author Contributions, Funding, Institutional Review Board Statement, Informed Consent Statement, Data Availability Statement, Acknowledgments, Conflicts of Interest.

Author Response

Reviewer 2

The article titled "OMICS in Human Milk: Focus on Biological Effects on Bone Homeostasis" explores the role of human milk (HM) in infant bone health through the lens of omics technologies, including proteomics, metabolomics, and transcriptomics. It discusses how specific HM components like proteins, lipids, hormones, and oligosaccharides contribute to bone metabolism, calcium absorption, and osteoblast differentiation, supporting infant skeletal growth. The article also highlights the role of maternal diet, genetic factors, and the infant's microbiome in shaping HM's composition and, consequently, its impact on bone health. Despite significant findings, the authors emphasize the need for longitudinal studies to fully understand the complex interactions between HM components and bone development, aiming to optimize early-life nutrition for better skeletal outcomes.

Below are some necessary changes.

Apply italics to all scientific names throughout the text.         

Done.

Line 40 – The keywords should be different from those used in the title, and it is recommended that they are not present in the abstract.  

We have revised the keywords as requested to avoid duplication with the title and abstract content.

Line 47 – "Microbes" or "microorganisms"?  

We have revised the term to "microorganisms" for clarity and consistency within the text.

Lines 104-106 – Does the age of the mother influence the nutrient quality of HM?  

Thank you for the question regarding the influence of maternal age on the nutrient quality of human milk (HM). Based on our review of the available literature, we did not find any studies that demonstrate a significant relationship between the mother’s age and the nutrient composition of HM. Factors such as maternal diet, health, and environmental influences appear to play more substantial roles in determining HM composition.

Lines 120-121 – Change vitamin B6 and vitamin B12 to pyridoxine and cyanocobalamin,

respectively, to be consistent with the names used for other B-complex vitamins.  

Done.

Section 2.1 – The authors addressed the importance of calcium, potassium, and vitamin D. What about the importance of other vitamins and minerals? Or are they not significant?  

Thank you for your valuable feedback. In response to your suggestion, we have enriched the discussion to include the significance of other essential vitamins and minerals present in human milk, such as magnesium, zinc, iodine, vitamins A and K, and their respective roles in supporting infant development.

Lines 315-316 – Does the legend refer to Figure 1?  

Yes, we have changed the font of the legend to make it clearer.

Line 366 – "...essential nutrients and bioactive compounds..." Up to this point, the authors have presented macro- and micronutrients but have not introduced bioactive compounds. It would be appropriate to include a section on this class of components.  

We have added a dedicated section on bioactive compounds to the manuscript. This addition now clarifies and expands upon the mention of "essential nutrients and bioactive compounds" within human milk. The new section outlines the distinct biological roles these compounds play, complementing the previously discussed macro- and micronutrients and providing a comprehensive understanding of the functional components in human milk.

Line 473 – Is there any relationship between adiponectin and insulin response?  

Thank you for your insightful comment regarding the relationship between adiponectin and insulin response. We have clarified this connection in the revised manuscript.

Figure 2 should be inserted right after it is mentioned in the text. 

Done.
